# Theseus:
# A Library for Differentiable Nonlinear Optimization

**Luis Pineda**[1], **Taosha Fan**[1], **Maurizio Monge**[2], **Shobha Venkataraman**[1],
**Paloma Sodhi**[1], **Ricky T. Q. Chen**[1], **Joseph Ortiz**[1], **Daniel DeTone**[2], **Austin Wang**[1],
**Stuart Anderson**[1], **Jing Dong**[2], **Brandon Amos**[1], **Mustafa Mukadam**[1]

[1]Meta AI, [2]Reality Labs Research

## Abstract

We present Theseus, an efficient application-agnostic open source library for differentiable nonlinear least squares (DNLS) optimization built on PyTorch, providing a common framework for end-to-end structured learning in robotics and vision. Existing DNLS implementations are application specific and do not always incorporate many ingredients important for efficiency. Theseus is application-agnostic, as we illustrate with several example applications that are built using the same underlying differentiable components, such as second-order optimizers, standard costs functions, and Lie groups. For efficiency, Theseus incorporates support for sparse solvers, automatic vectorization, batching, GPU acceleration, and gradient computation with implicit differentiation and direct loss minimization. We do extensive performance evaluation in a set of applications, demonstrating significant efficiency gains and better scalability when these features are incorporated. Project page: https://sites.google.com/view/theseus-ai/

## 1 Introduction

Reconciling traditional approaches with deep learning to leverage their complementary strengths is a common thread in a large body of recent work in robotics. In particular, an emerging trend is to differentiate through nonlinear least squares (NLS) [1] which is a second-order optimization formulation at the heart of many problems in robotics [2–7] and vision [8–13]. Optimization layers as inductive priors in neural models have been explored in machine learning with convex optimization [14, 15] and in meta learning with gradient descent [16, 17] based first-order optimization.

Differentiable nonlinear least squares (DNLS) provides a general scheme to encode inductive priors, as the objective function can be partly parameterized by neural models and partly with engineered domain-specific differentiable models. Here, as illustrated in Fig. 1, input tensors define a sum of weighted squares objective function and output tensors are minima of that objective. Such implicit layers [18] are in contrast to typical (explicit) layers that take input tensors through a linear transformation and some element-wise nonlinear activation function.

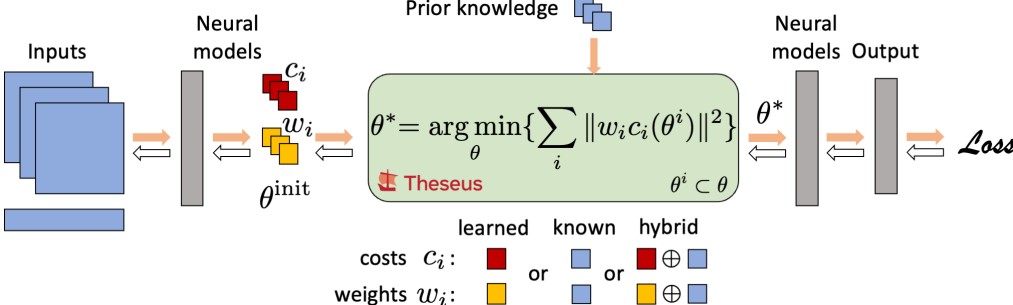

Figure 1: Theseus enables building custom, efficient DNLS layers that support end-to-end structured learning.

36th Conference on Neural Information Processing Systems (NeurIPS 2022).

The ability to compute gradients end-to-end is retained by differentiating through the optimizer which allows neural models to train on the final task loss, while also taking advantage of priors captured by the optimizer. The flexibility of such a scheme has led to promising state-of-the-art results in a wide range of applications such as structure from motion [19], motion planning [20], SLAM [21, 22], bundle adjustment [23], state estimation [24, 25], image alignment [26] with other applications like manipulation and tactile sensing [27, 28], control [29], human pose tracking [30, 31] to be explored. However, existing implementations from above are application specific, common underlying tools like optimizers get reimplemented, and features like sparse solvers, batching, and GPU support that impact efficiency are not always included. This has led to a fragmented literature where it is difficult to start work on new ideas or to build on the progress of prior work.

To address this gap, we present Theseus, an open source library for differentiable nonlinear least squares optimization built on PyTorch. Theseus provides an efficient application-agnostic interface that consolidates recent efforts and catalyzes future progress in the domain of structured end-to-end learning for robotics and vision. Our contributions are summarized below.

**Application agnostic interface.** Our implementation provides an easy to use interface to build custom optimization layers and plug them into any neural architecture. (i) The layer can be constructed from a set of available second-order optimizers like Gauss-Newton, Levenberg–Marquardt (with adaptive damping) and Dogleg, and a nonlinear least squares objective. (ii) The objective can be constructed with learnable or hand-specified cost functions, either by applying one of many common costs already provided in the library, or by building custom costs in-place with support for automatic differentiation through PyTorch [32]. (iii) We also provide differentiable Lie groups for representing 2D/3D positions and rotations [33], and differentiable kinematics wrapping over an existing library [34] for representing robot models. More details are described in Sec. 3.

**Efficiency based design.** Efficiency is a central design consideration and we make several advancements in improving computation times and memory consumption. (i) As common in prior work, an optimizer implementation using PyTorch's native linear solver would use a dense representation for solving the linear system within the nonlinear optimization. In practice, these optimization problems often have a considerable amount of sparsity that can be exploited [35–38]. In Theseus, we implement sparse linear solvers that are differentiable end-to-end and make them efficient with custom CPU and CUDA backends to support batching and GPU acceleration. (ii) Beyond sparse solvers, we extend batching and GPU support to all features in the library and add automatic vectorization of cost functions and other operations to significantly boost efficiency. (iii) Finally, we introduce implicit differentiation [39] and direct loss minimization [40, 41], which have been previously applied to only first order optimizers like gradient descent and convex optimization, to a new class of second-order optimizers. This goes beyond prior work with nonlinear least squares that currently only support differentiation with standard unrolling, which is known to have challenges with compute, memory, and vanishing gradients. More details are described in Sec. 4.

**Highlights of results.** Together, the application-agnostic features let users easily set up a variety of problems like pose graph optimization, tactile state estimation, bundle adjustment, motion planning, and homography estimation, all of which are included as examples in the open source code and described in Sec. 3.1. In evaluations, we find that on a standard GPU, Theseus with a sparse solver is much faster and requires significantly less memory than a dense solver, and when solving a batch of large problems the forward pass of Theseus is up to 20x faster than state-of-the-art C++ based solver Ceres that has limited GPU support and does not support batching and end-to-end learning. We also compare all backward modes to find that with increasing number of optimization iterations, compute and memory increases linearly for unrolling and stays constant for implicit differentiation, while the latter also provides better gradients. More details are described in Sec. 5.

## 2 Background and related work

**Nonlinear least squares (NLS)** is an optimization problem [1] that finds optimization variables $\theta$

$$\theta^\star = \arg\min_\theta S(\theta), \qquad S(\theta) = \frac{1}{2} \sum_i ||r_i(\theta^i)||^2 = \frac{1}{2} \sum_i ||w_i c_i(\theta^i)||^2 \qquad (1)$$

where the objective $S(\theta)$ is a sum of squared vector-valued residual terms $r_i$, each a function of $\theta^i \subset \theta$ that are (non-disjoint) subsets of the optimization variables $\theta = \{\theta_j\}$. Any variable $\theta_j$ is a manifold object; for example, a Euclidean vector or a matrix Lie group. For flexibility, we represent a residual $r_i(\theta^i) = w_i c_i(\theta^i)$ as a product of a matrix weight $w_i$ and vector cost $c_i$. **Robotics and**

**vision** have used this general optimization formulation to tackle many applications [3, 4]. For example, costs capture sensor measurement errors and physical constraints to optimize camera, robot, object, or human poses in estimation and tracking problems like simultaneous localization and mapping (SLAM) [42], structure from motion [13], bundle adjustment [8], visual inertial odometry [2], articulated tracking [12], contact odometry in legged locomotion [27], 3D pose and shape reconstruction of humans [30, 31] or objects [10]. Similarly, costs can also capture constraints and desired future goals to find robot states or actions in motion planning [5], dynamics [6], and control [29] problems.

**Solving NLS.** Problems represented by Eq. (1) are solved by iteratively linearizing the nonlinear objective around the current variables to get the linear system $(\sum_i J_i^\top J_i)\delta\theta = (\sum_i J_i^\top r_i)$, then solving the linear system to find the update $\delta\theta$, and finally updating the variables $\theta \leftarrow \theta - \delta\theta$, until convergence. Note that in the update the minus operation is more generally a retraction mapping for non-Euclidean variables. In the linear system, $J_i = [\partial r_i/\partial\theta^i]$ are the Jacobians of residuals with respect to the variables and the iterative method above, called Gauss-Newton (GN), is a nonlinear optimizer that is (approximately) second-order, since $H = (\sum_i J_i^\top J_i)$ represents the approximate Hessian. To improve robustness and convergence, variations like Levenberg–Marquardt (LM) damp the linear system, while others use a trust region and adjust step size for the update with line search (e.g., Dogleg). Please refer to [1, 43] for an in-depth exploration. In most applications discussed above the objective structure gives rise to a sparse Hessian, since not all costs depend on all variables. Several general purpose frameworks [35–38] have been built that leverage this sparsity property to efficiently solve the sparse linear system in every iteration of the nonlinear optimization. While these frameworks were not built for deep learning, they are highly efficient and performant on CPU.

**NLS with learning.** Data driven learning has been explored to address challenges in hand crafting costs or features for costs, finding weights to balance different costs, or to find initializations that lead to better convergence. Some examples include, learning object shape code [44] or environment depth code [45] for SLAM [46], learning motion priors for planning to manipulate articulated objects [47], learning relative pose from tactile images to estimate object state during pushing [28], and semantic 2D segmentation fused in 3D mesh for semantic SLAM [7]. These approaches only train features on a surrogate or intermediate loss and then apply optimization at inference where the true downstream task loss is available but not utilized. To take full advantage of end-to-end learning, latest approaches thus are redesigning the optimization to be differentiable.

**Differentiable NLS (DNLS)** solves the optimization in Eq. (1) and also provides gradients of the solution $\theta^\star$ with respect to any upstream neural model parameters $\phi$ that parameterize the objective $S(\theta; \phi)$ and in turn any costs $c_i(\theta^i; \phi)$, weights $w_i(\phi)$, or initialization for variables $\theta_{init}(\phi)$. The goal is to learn these parameters $\phi$ end-to-end with a downstream learning objective $L$ defined as a function of $\theta^\star$. This results in a bilevel optimization setup as shown in Fig. 1

$$\text{inner loop: } \theta^\star(\phi) = \arg\min_\theta S(\theta; \phi), \qquad \text{outer loop: } \phi^\star = \arg\min_\phi L(\theta^*(\phi)) \qquad (2)$$

where the inner loop is DNLS and the outer loop is gradient descent class of optimization that is standard in deep learning. The outer loop performs update $\phi \leftarrow \phi + \delta\phi$ by computing $\delta\phi$ using gradients $\partial\theta^\star/\partial\phi$ through inner loop DNLS. Note that more generally the learning objective i.e. outer loss $L$ can also depend on other quantities like neural model parameters downstream of $\theta^*$, but we omit them here for clarity.

**Recent works with DNLS** have outperformed optimization only or learning only methods by combining the strengths of classical methods with deep learning. For example, learning features for costs to represent depth in bundle adjustment [23] and monocular stereo [48] where an initialization network also learns to predict depth and pose, learning cost weights like motion model weights in video to depth estimation [19], obstacle avoidance weights in 2D motion planning from occupancy images [20], learning robust loss weights in image alignment [26] and state-of-the-art dense SLAM [22], and confidence weights for feature matching to optimize camera pose [49]. Other works, backpropagate reconstruction error to sensor model in a SLAM system [21], solve large scale bundle adjustment on a GPU [50], and learn sensor and dynamics models for 2D visual object tracking and visual odometry [24]. These implementations however, are application specific which has led to repeated work in building DNLS where features like learnable costs and weights, Lie groups, and kinematics are not always present. Additionally, features that have a significant impact on performance, like sparsity and vectorization of costs are only considered by some [24, 50, 51] or in the case of implicit differentiation for NLS optimization, have not yet been explored.

# 3 Application agnostic interface

Given the lack of a common and efficient framework for DNLS an important goal of Theseus is to provide an application-agnostic interface. In this section, we describe how we enable this with an easy-to-use core API, standard cost functions, and features like Lie groups and kinematics, and illustrate several examples using this interface. We discuss design for efficiency in the next section.

```python
1   x_true, y_true, v_true = read_data() # shapes (1, N), (1, N), (1, 1)
2
3   x = th.Variable(torch.randn_like(x_true), name="x")
4   y = th.Variable(y_true, name="y")
5   v = th.Vector(1, name="v") # a manifold subclass of Variable for optim_vars
6
7   def error_fn(optim_vars, aux_vars):  # returns y - v * exp(x)
8       x, y = aux_vars
9       return y.tensor - optim_vars[0].tensor * torch.exp(x.tensor)
10
11  objective = th.Objective()
12  cost_function = th.AutoDiffCostFunction(
13      [v], error_fn, y_true.shape[1], aux_vars=[x, y],
14      cost_weight=th.ScaleCostWeight(1.0))
15  objective.add(cost_function)
16  layer = th.TheseusLayer(th.GaussNewton(objective, max_iterations=10))
17
18  phi = torch.nn.Parameter(x_true + 0.1 * torch.ones_like(x_true))
19  outer_optimizer = torch.optim.Adam([phi], lr=0.001)
20  for epoch in range(10):
21      solution, info = layer.forward(
22          input_tensors={"x": phi.clone(), "v": torch.ones(1, 1)},
23          optimizer_kwargs={"backward_mode": "implicit"})
24      outer_loss = torch.nn.functional.mse_loss(solution["v"], v_true)
25      outer_loss.backward()
26      outer_optimizer.step()
```

Listing 1: Simple DNLS example with Theseus, see App. B for details.

The core API lets users focus on describing the DNLS problem and their interaction with the outer loss $L$ and parameters $\phi$ within any broader PyTorch model, while the solution and differentiation are seamlessly taken care of under-the-hood. The basic components of the core API are described below with the help of a simple example in Listing 1 (see App. B for more details on the example):

- Variable: refers to either *optimization variables*, $\theta$, or *auxiliary variables* (those constant with respect to $S$, e.g., parameters $\phi$ or data tensors), which are named wrappers of torch batched tensors stored in Variable.tensor (lines 3-5).
- CostFunction: defines costs $c_i$ (lines 12-14) and are also responsible for declaring which of its variables are optimization and which are auxiliary (lines 8-9),
- CostWeight: defines weights $w_i$ associated with cost $c_i$ (line 14).
- Objective: defines $S(\theta; \phi)$, and thus the structure of an optimization problem (lines 11, 15) by holding all cost functions and weights, and their associated variables. These are implicitly obtained when a CostFunction is added to the Objective, and Variable names are used to infer which are shared by one or more CostFunction.
- Optimizer: is the inner loop optimization algorithm (e.g. Gauss-Newton) that finds the solution $\theta^\star$ given objective $S$ (line 16).
- TheseusLayer: encapsulates Optimizer and Objective, and serves as the interface between the DNLS block and other torch modules upstream or downstream (line 16).

The interface between the inner loop optimization and the outer loop's parameters and loss occurs via TheseusLayer.forward (lines 21-23). This receives as input a dictionary mapping variable names to torch tensors, which Theseus then uses to populate the corresponding Variable with the tensor mapped to its name. With the input dictionary users can provide initial values for the optimization variables, data tensors, or current values for parameters $\phi$ before running the inner loop optimization. The output of forward is another dictionary that maps variable names to tensors with their optimal values found in the inner loop (lines 21, 24); auxiliary variables are not modified during the forward

pass. The output tensors can then be combined with other `torch` modules downstream to compute $L$, while maintaining the full differentiable computation graph (lines 24-26).

We currently provide Gauss-Newton, Levenberg–Marquardt (with adaptive damping), and Dogleg as nonlinear `Optimizer` for the inner loop, with the ability to easily add support for more optimizers in the future. Listing 1 uses `AutoDiffCostFunction` to construct an in-place `CostFunction` (line 12) which allows automatically calculating Jacobians $J_i$ with PyTorch (see App. C). Beyond this, in the library we include standard cost functions with analytical Jacobians broadly used in many applications, like Gaussian measurements, reprojection error, relative pose measurement, motion models, and collision costs. We also include a variety of robust loss functions, useful for example in handling outliers [52], which can be easily integrated with `CostFunction`. Next we describe support for Lie groups and kinematics.

**Differentiable Lie groups.** Lie groups are widely used in robotics and vision to represent 2D/3D positions and rotations [33]. Due to their non-Euclidean geometry, it is difficult to apply them to deep learning, which primarily operates with Euclidean tensors, but recently there is growing interest in making them compatible [24, 53–57]. LieTorch [54] generalizes automatic differentiation on the Lie group tangent space through local parameterization around the identity, but the implementation is complex since every operation requires a custom kernel. In contrast, `Theseus` computes common Lie group operators, e.g., the exponential and logarithm map, inverse, composition, etc., in closed form, and provides their corresponding analytical derivatives on the tangent space. Following [58], we also implement a projection operator that allows us to project gradients computed by `PyTorch`'s autodiff to the tangent space and use them to easily compute Jacobians and update Lie group variables correctly; a similar strategy has also been implemented in [59]. Additionally, our Lie group implementation includes a heuristic extension that allows using any of `PyTorch`'s first-order optimizers on non-Euclidean manifolds with minimal code changes. All of these make it easy and straightforward to run optimization and train neural networks with Lie groups variables. More details in App. D.

**Differentiable kinematics.** Many problems such as motion planning or state estimation on high degree of freedom robots like arms or mobile manipulators, involve computation of robot kinematics for collision avoidance or computing distance of end effector to goal. `Theseus` provides a differentiable implementation of forward kinematics by wrapping over Differentiable Robot Model [34], which builds a differentiable kinematics function from a standard robot model file. Gradients are computed through autodiff, while we also provide a more efficient, analytical manipulator Jacobian. This module can be used within any `CostFunction` in `Theseus`.

## 3.1 Example applications

To illustrate the versatility of `Theseus`, we include a number of example DNLS applications below with more details in App. E. Crucially, to implement these with `Theseus`, most of the effort is only in defining application-specific components such as data management, neural models, or custom `CostFunction`. With these defined, putting the full DNLS block together is a few lines of code to setup a `TheseusLayer` and an outer loop, similar to the simple example in Listing 1.

**Pose graph optimization (PGO)** estimates poses from their noisy relative measurements [60]. With DNLS we learn the radius of a Welsh robust cost function for outlier rejection, using the difference between estimated and ground truth poses as the outer loss on a synthetic dataset.

**Tactile state estimation** follows [28], which estimates 2D poses of an object pushed by a robot hand with an image-based tactile sensor [61]. A neural network that predicts relative pose between hand and object from tactile images is learned end-to-end through the `TheseusLayer`.

**Bundle adjustment** is the problem of optimizing a 3D reconstruction formed by a set of camera images and a set of landmarks observed and matched across the images [62]. We learn the radius of a soft-kernel that penalizes outlier observations, using the average frame pose error as outer loss.

**Motion planning** considers a differentiable version of the GPMP2 planning algorithm, inspired by [20], where the outer loss tries to match expert demonstrations. Here we learn a model for initializing optimization variables, and we include the inner loop objective as a term in the outer loss.

**Homography estimation.** Homography is a linear transformation between corresponding points in two images and can be solved by minimising a dense photometric loss. Robustness to lighting and viewpoint change can be improved with a feature-metric loss based on CNN features [63–68]. In our outer loop, we train a CNN to produce robust features for image alignment.

## 4 Efficiency based design

Theseus enables several different applications with a general interface. Compute and memory efficiency are central to making its usage practical. Next, we explain design considerations to support batching and vectorization, sparsity, and backward modes for differentiation, which we demonstrate boost performance in the evaluations section.

### 4.1 Batching and vectorization

Parallel processing is important to improve computational efficiency in machine learning and optimization. In Theseus, we enable two levels of parallelization. First, Theseus natively supports solving a batch of DNLS in parallel, thus fitting seamlessly in the PyTorch framework, where training and inferences on batches is the standard. Second, inspired by DeepLM [50], and noting that lots of the operations such as costs, gradients/Jacobian computation, and variable

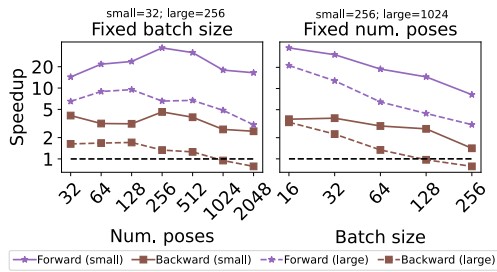

Figure 2: Speedup with automatic vectorization on PGO. Black dotted line is without vectorization.

updates only differ from each other in terms of the input data, we make use of the single-instruction-multiple-data (SIMD) protocol by automatically detecting and vectorizing operations of the same type, significantly reducing computation overhead. Using the PGO example, Fig. 2 shows that Theseus achieves significant speedup with automatic vectorization both for forward and backward pass. Note that there is an application-dependent trade-off between memory and speed; here the memory use increases by up to $\sim 82\%$ for forward and $\sim 55\%$ for backward.

### 4.2 Handling sparsity with linear solvers beyond PyTorch

Solving NLS requires solving a sequence of linear systems to obtain descent directions. As discussed in Sec. 2, these systems are generally sparse and can be solved much more efficiently if not treated as dense. Theseus includes differentiable sparse solvers that take advantage of the sparsity, complementing PyTorch's native dense solvers. Importantly, Theseus seamlessly takes care of assembling the cost functions and variables in the objective into sparse data structures that our linear solvers can consume, without any extra burden on the user. Currently, we provide three sparse solvers: (i) a CPU-based solver that relies on CHOLMOD [69], (ii) cudaLU, which is based on the cuSolverRF package that is part of Nvidia's cuSolver library provided with CUDA, and (iii) BaSpaCho, our novel batched sparse Cholesky solver with GPU support. As a bonus feature, we provide access to these solvers as standalone PyTorch functions, so they can be used to solve sparse matrices arising outside of NLS or DNLS optimization.

**CHOLMOD-based solver.** CHOLMOD [69] achieves state-of-the-art performance on computation of the Cholesky decomposition of sparse matrices. It exploits parallelism by grouping sparse entries to take advantage of high-performance multi-threaded dense matrix operations in BLAS/LAPACK libraries. CHOLMOD has some limited support for GPU for some of its operations, but the algorithm is strongly CPU-based, and the user is expected to provide matrix data on the CPU. One convenient feature is computing the symbolic analysis of a sparse matrix pattern as a separate step and creating a symbolic decomposition object that can be used for all subsequent factorizations. We also take advantage of builtin functionality for sparse multiplication and only provide the Jacobian matrix $J$ to solve for the Hessian $H = J^\top J$. Two limitations of the library with respect to Theseus are, first, the lack of proper GPU support, which forces us to provide matrix data on the CPU, and, second, the lack of batching, which requires us to loop to solve every problem in the batch independently. On the other hand, since it runs on CPU, it has less memory restrictions than GPU-based solvers (see Sec. 5.1).

**cudaLU solver.** cuSolverRF is designed to accelerate the solution of sets of linear systems by fast LU refactorization when given new coefficients for the same sparsity pattern. To take advantage of this, we implemented custom CUDA kernels for batched sparse matrix-matrix and matrix-vector products, and for solving a batch of sparse linear systems using LU factorization from cuSolverRF. Although this solver leads to a substantial performance boost over PyTorch's dense solver (see Sec. 5.1), the closed-source nature of cuSOLVER results in some challenges and limitations: (i) cuSolverRF does not support separate symbolic decomposition and numeric contexts, so it's not possible to use the same symbolic decomposition to hold in memory separate factors. Since this is necessary in Theseus for unrolling of the inner loop, we work around this limitation by creating a pool of contexts, and we use the least recently used context for factorization. As a consequence, the number of contexts must

be set according to the number of iterations that need to be unrolled; (ii) The batch size is fixed once a context is created. Since recreating the contexts is an expensive operation, it means that the batch size has to be constant over the course of outer loop optimization; (iii) It relies on LU factorization, which for symmetric matrices (the case of `Theseus`) is less efficient than using Cholesky decomposition.

**BaSpaCho solver.** Batched Sparse Cholesky (BaSpaCho) is a novel open-source sparse Cholesky solver designed for `Theseus` with support for batching ([https://github.com/facebookresearch/baspacho](https://github.com/facebookresearch/baspacho)). BaSpaCho implements the *supernodal* Cholesky algorithm [70] to achieve state-of-the art performance by exploiting dense operations via BLAS/cuBLAS. This is achieved by building an elimination tree and then clustering column blocks with similar sparsity patterns. These blocks form nodes of the elimination tree and allow dense operations. In BaSpaCho, the dense operations are dispatched to BLAS (on CPU) or cuBLAS (on GPU), with additional support added on top for batching matrix operations with the same sparsity patterns. In problems with very sparse matrices, like bundle adjustment [8], the supernodal algorithm employed in state-of-the-art solvers [37] is unable to eliminate columns of parameter blocks simultaneously. Thus, past work has resorted to the Schur complement trick [71] to send a reduced problem to the sparse solver. However, this logic adds extra complexity to the nonlinear optimization, while essentially duplicating the work of the (mathematically equivalent) Cholesky decomposition. In BaSpaCho, we instead complement the supernodal algorithm with *sparse elimination* that removes the need to externally handle Schur complement as a workaround to the limitation of the supernodal algorithm. More details are described in App. F.

**Backward for custom linear solvers.** Obtaining derivatives of the linear system solve with respect to the parameters is a crucial operation for DNLS. In particular, we consider optimizing the parameters $A$ and $b$ of a linear system $y = A^{-1}b$ to minimize a downstream function $f(y)$. The derivatives of the loss with respect to the parameters of the linear system can be obtained with implicit differentiation, $\frac{\partial f}{\partial b} = A^{-1}\frac{\partial f}{\partial y}$ and $\frac{\partial f}{\partial A} = -A^{-1}\frac{\partial f}{\partial y}y^\top$, as done in Barron and Poole [72]. In `Theseus`, we implement this by connecting the Python interface of our sparse solvers with PyTorch's `autograd.Function` classes that implement the gradients above in their `backward` methods. This connects the computation graph between the downstream function and any upstream parameters that modify the system via auxiliary variables or values for optimization variables. Furthermore, since the gradients require solving linear systems that use the same matrix as the forward pass, our backward pass can cache factorizations, resulting in it being significantly faster than the forward pass (see Fig. 3).

### 4.3 Backward modes for DNLS

The parameters $\phi$ upstream of DNLS can be learned end-to-end through the solution $\theta^\star(\phi)$ by using the *adjoint derivatives* $\partial\theta^\star(\phi)/\partial\phi$. We include four methods for computing them in `Theseus`.

**Unrolling** is the standard way in which past work in DNLS has computed the adjoint derivatives. This is often referred to as backpropagation through time or unrolled optimization and is explored in [16, 20, 73–82]. In practice, often only a few steps of unrolling are performed due to challenges with compute, memory, and vanishing gradients.

**Truncated differentiation.** Aside from unrolling a few steps, another way of approximating the derivatives is to use truncated backpropagation through time (TBPTT) [83, 84]. Truncation unfortunately results in biased derivatives and many works [85–89] seek to further theoretically understand the properties of TBPTT, including the bias of the estimator and how to unbias it.

**Implicit differentiation.** If $\theta^\star$ can be computed exactly, then the implicit function theorem provides a way of computing the adjoint derivatives, as done in related work in convex optimization and first-order gradient descent methods [14, 15, 90–95]. We apply the implicit function theorem from Dontchev and Rockafellar [39, Theorem 1B.1] (see App. H) to Eq. (2) to perform implicit differentiation on a new class of second-order NLS optimization. This first requires that we transform Eq. (2) into an implicit function that finds the roots. We do this via the first-order optimality condition, resulting in $g(\theta;\phi) := \nabla_\theta S(\theta;\phi)$. Finding $\Theta^\star(\phi) := \{\theta \mid g(\theta;\phi) = 0\}$ corresponds to solving Eq. (2). Under mild assumptions, the theorem above gives the adjoint derivative at $\bar\phi$

$$\mathrm{D}_\phi\theta^\star(\bar\phi) = -\mathrm{D}_\theta^{-1}g(\theta^\star(\bar\phi);\bar\phi)\mathrm{D}_\phi g(\theta^\star(\bar\phi);\bar\phi). \tag{3}$$

As `Theseus` internally uses a (Gauss-)Newton solver, the following proposition provided in App. H shows how we can compute Eq. (3) by differentiating a single Newton step at an optimal solution.

**Proposition 1.** *The implicit derivative (Eq. (3)) can be computed by differentiating a Newton step* $h(\theta;\phi) := \theta - [\nabla_\theta^2 S(\theta;\phi)]_{\mathrm{stop}}^{-1}\nabla_\theta S(\theta;\phi)$ *at an optimal* $\theta^\star$, *where* $[\cdot]_{\mathrm{stop}}$ *zeros the derivative.*

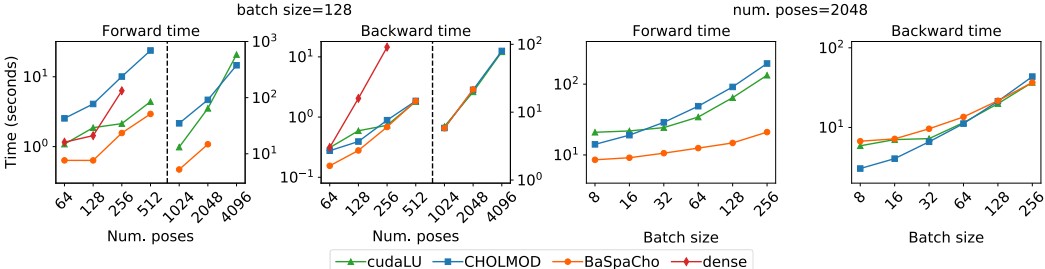

Figure 3: Forward/backward times of Theseus with sparse and dense solvers on different PGO problem scales.

**Direct loss minimization.** Suppose we have an outer loss as in Eq. (2). The direct loss minimization (DLM) approach uses this loss to augment the inner-loop optimization problem in order to define a finite difference scheme that approaches the true gradient $\nabla_\phi L = \lim_{\varepsilon \to 0} g^\varepsilon_{\text{DLM}}$, where $g^\varepsilon_{\text{DLM}} \triangleq \frac{1}{\varepsilon}\{\frac{\partial}{\partial \phi}S(\theta^*; \phi) - \frac{\partial}{\partial \phi}S(\theta_{\text{direct}}; \phi)\}$. This was used in prior works that solve optimization problems on structured discrete domains [40, 41, 96, 97], but has so far not seen much use in structured continuous settings. We modify the original DLM formulation to better suit its implementation within Theseus

$$\theta^\star = \arg\min_{\hat{\theta}} S(\hat{\theta}; \phi), \qquad \theta_{\text{direct}} = \arg\min_{\hat{\theta}} S(\hat{\theta}; \phi) + \left\| \varepsilon\hat{\theta} - \tfrac{1}{2}\nabla_\theta L(\theta^*) \right\|^2. \tag{4}$$

This is different from the original formulation in two ways: (i) we only assume access to the gradient vector $\nabla_\theta L(\theta^*)$, which helps formulate DLM as an algorithm for computing vector-Jacobian products, and (ii) we add a small regularization term to ensure the modified objective for $\theta_{\text{direct}}$ is a sum-of-squares without affecting the limit as $\varepsilon \to 0$. See App. H for more details.

## 5    Evaluation

We evaluate the performance of Theseus under different settings with PGO and tactile state estimation applications from Sec. 3.1. PGO allows us to easily control the problem scales for performance evaluation; in Sec. 5.1 we profile time and memory consumption of Theseus in an end-to-end setup and in Sec. 5.2 we evaluate timings of Theseus as a stand-alone NLS optimizer and compare with state-of-the-art Ceres [37]. The tactile state estimation application involves a more complex outer loop model that is useful for comparing all different backward modes, which we present in Sec. 5.3.

### 5.1    Profiling forward and backward pass of Theseus for DNLS

We study the performance of Theseus for DNLS on the PGO problem [60] with the synthetic Cube dataset, as described in App. E. We run 10 inner loop iterations and 20 outer loop epochs, and use implicit differentiation to compute gradients of the inner NLS optimization. For these experiments we used an Nvidia V100 GPU with 32GBs of memory for all Python computation, and Intel Xeon 2.2GHz CPU with 20 threads for the CPU-based CHOLMOD linear solver. We evaluate performance using our sparse solvers in Theseus and using PyTorch's Cholesky dense solver.

Fig. 3 shows the average time of a full forward and backward pass for a given batch size, taken by Theseus with different solvers (cudaLU, CHOLMOD, BaSpaCho and dense), for different problem scales (number of poses and batch size). The two left plots show time as a function of number of poses for a batch size of 128, while the two right plots show time as a function of batch size for 2048 poses. We find that dense does not scale well with poses or batch size. For a batch size of 128, the largest problem that it can solve before running out of GPU memory has 256 poses (left two plots). With 2048 poses, dense is unable to solve the problem regardless of batch size (right two plots). On the other hand with a batch size of 128, our solvers BaSpaCho scale to 2048 poses and cudaLU scale to 4096 poses. CHOLMOD can solve problems even larger, since the linear system is solved on CPU and we have successfully tested up to 8192 poses and batch size 256 (see App. G), for a total of 22GBs of GPU usage for residuals and Jacobian blocks computation.

In addition to being more memory efficient, running times of our sparse solvers are also smaller for large enough number of poses/batch size, especially for the backward pass. Even though dense's total time for forward+backward is comparable to cudaLU and faster than CHOLMOD for smaller problems: e.g., 1.47s (dense) vs. 1.32s (cudaLU) and 2.82s (CHOLMOD) for batch size 128 and 64 poses, dense is significantly slower or out of memory for larger problems. For the largest problem that dense can solve (batch size 128 and 256 poses) its total time is already much slower than all others methods:

20.81s (dense) vs. 10.96s (CHOLMOD), 2.86s (cudaLU), and 2.25s (BaSpaCho). Furthermore, BaSpaCho outperforms dense for any problem scale and is up to one order of magnitude faster, including for smaller batch sizes and number of poses (see App. G for more results and details). For the largest problem that we consider (batch size 256 and 2048 poses), the total times for our sparse solvers are 170.28s for cudaLU, 239.07s for CHOLMOD, and 57.67s for BaSpaCho.

## 5.2 Profiling Theseus as stand-alone NLS optimizer

DNLS typically involves solving numerous optimization problems each epoch where a fast NLS optimizer is essential. We compare Theseus as a stand-alone NLS optimizer with the state-of-the-art Ceres [37] library for solving a batch of PGO problems without any learning involved. We compare all solvers in terms of the total time required to perform 10 iterations on a set of 256 PGO problems. CPU/GPU configurations are same as before. For CHOLMOD, we also include a configuration that runs everything on CPU, including Jacobians and residual computation (labelled CHOLMOD-allcpu).

Fig. 4 shows speedup obtained by Theseus with batching, vectorization and sparse solvers, over Ceres as a function of increasing number of poses or batch size. We vary the number of poses for a fixed batch size of 256, and vary the batch size for a fixed number of poses of 2048. Although Ceres is faster than all of our solvers when the number of poses and batch size are small (for instance, Ceres is 25x faster with 256 poses and 16 batch size, see App. G), as these increase Theseus shows significant speedup by being able to solve larger batches of problems in parallel. For the largest setup that

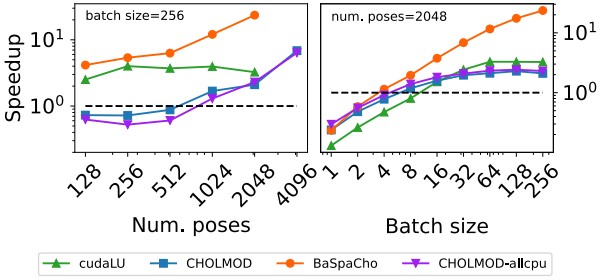

Figure 4: Speedup of Theseus (forward pass) over Ceres (black dashed) on different PGO problem scales.

all our solvers can scale to (2048 poses, 256 batch size), BaSpaCho is ∼23x faster than Ceres, and our other solvers are ∼4x faster. CHOLMOD has a 6x speedup for its largest setting (4096 poses, 256 batch size).

Since typical use case of Theseus involves large batches and number of variables during end-to-end learning with DNLS, the speedups in this setting against a performant NLS solver highlights the significance of our efficiency-based design choices. See App. G for additional results of smaller fixed batch size and number of poses.

## 5.3 Backward modes analysis

We explore the trade-offs between our different backward modes using the tactile state estimation application in Sec. 3.1. The learnable components here include a neural network, and thus closely follow the type of applications that motivate Theseus. We compare the following backward modes: derivative unrolling (Unroll), implicit differentiation (Implicit), truncated differentiation (Trunc), and direct loss minimization (DLM); for Trunc we include results when truncating 5 and 10 steps. We compare all modes along 3 axis of performance: validation loss after 100 epochs (outer loop), run time during training, and peak GPU memory consumption of TheseusLayer. For these experiments we used Quadro GP100 GPUs with 16GB of memory. For time and memory we present separate results for forward and backward pass, and all numbers are averaged over 700 (7 batches for 100 epochs). Below we discuss our main findings from this analysis, and more results and details can be found in App. H.

Fig. 5 shows average run times for all backward modes as a function of the maximum number of iterations in the inner loop optimization. We observe that the time used in the forward pass (Fig. 5, far left) increases roughly linearly for all modes, all having similar times except for Unroll, which is slower than other modes. On the other hand, we observe stark differences in the backward pass time (Fig. 5, center left), where Unroll is the only method that has a linear dependence on the number of inner loop iterations. All other methods have a constant footprint for computing derivatives, independent of the number of inner loop iterations. As expected, increasing the number of iterations through which we backprop (5 or 10 for Trunc, all iterations for Unroll) increases the time necessary for a backward pass (Implicit = DLM < Trunc-5 < Trunc-10 << Unroll).

Figure 5 (center right) shows the average peak memory consumption of the backward modes. In this case, the trends observed for the backward pass memory consumption is similar to the trends

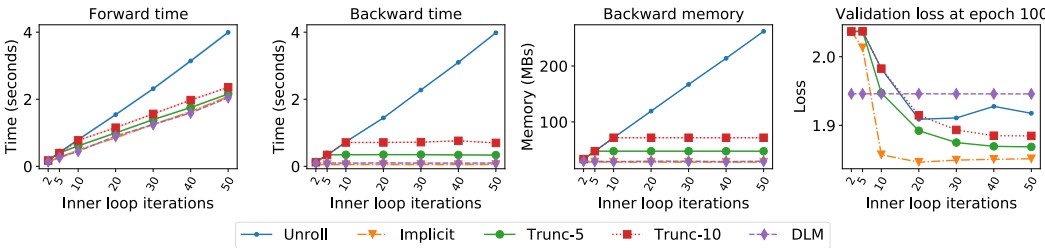

Figure 5: Time and memory consumption of different backward modes in tactile state estimation.

in time. In particular, `Unroll`'s memory footprint increases linearly with the number of inner loop iterations, from $\sim 34$MBs to $\sim 262$MBs; for all other methods the memory consumption remains constant. The best memory profiles in this example is obtained with `Implicit` and `DLM` backward modes, with $\sim 28$MBs and $\sim 29$MBs, respectively. These trends also hold for the forward pass memory consumption.

Figure 5 also shows the validation losses obtained with all backward modes (far right). The best validation loss, after 100 epochs of training, is obtained using `Implicit`, followed by `Trunc` variants. We notice that both variants of `Trunc` keep improving with increasing number of inner loop iterations, and that `Unroll` and `Implicit` achieve the best results with 20 iterations. One exception is `DLM`, which doesn't improve much with the number of iterations, but is also the best method when only 2 inner loop iterations are performed. As a point of caution, we stress that, unlike the timing and memory results, the relative training performance between different backward modes is likely to be application dependent, and is affected by hyperparameters such as the step size used for the inner loop optimizer (0.05 in this example), and the outer optimizer's learning rate. Our experiments suggest that implicit differentiation is a good default to use for differentiable optimization, considering its low time/memory footprint, and potential for better end-to-end performance with proper hyperparameter tuning.

## 6 Discussion

**Summary.** `Theseus` provides nonlinear least squares as a differentiable layer and enables easily building and training end-to-end architectures for robotics and vision applications. We illustrate several example applications using the same application-agnostic interface and demonstrate significant improvements in performance with our efficiency-based design. Following how autodiff and GPU acceleration (among others) have led to the evolution of `PyTorch` in contrast to `NumPy` [98], we can similarly view sparsity and implicit differentiation on top of autodiff and GPU acceleration as the key ingredients that power `Theseus`, in contrast to solvers like `Ceres` that typically only support sparsity. When solving a batch of large problems the forward pass of Theseus is up to 20x faster than `Ceres`.

**Limitations.** `Theseus` currently has a few limitations. The nonlinear solvers we currently support apply constraints in a soft manner (i.e., using weighted costs). Hard constraints can be handled with methods like augmented Lagrangian or sequential quadratic programs [99, 100], and differentiating through them are active research topics. The current implementation of LM does not support damping to be learnable. Some limitations and trade-offs with the sparse linear solvers are discussed in Sec. 4.2, and with backward modes are discussed in App. H. Online learning applications may require frequently editing the objective and depending on the problem size there may be a nontrivial overhead that is not currently optimized as we explored only non-incremental settings in this work. Additional performance gains can be extracted by moving some of our Python implementation to C++ but we prioritized flexibility in evolving the API in the short-term. We do not yet support distributed training beyond what `PyTorch` natively supports. We will explore these features and optimizations in the future as the library continues to evolve.

## Acknowledgments and Disclosure of Funding

The authors would like to thank Dhruv Batra, Olivier Delalleau, Jessica Hodgins, and Mary Williamson for guidance and support on the project, Dhruv Batra and Sal Candido for feedback on early drafts of the paper, Franziska Meier for help with the differentiable robot model library, Paul-Edouard Sarlin for helpful discussion on the homography example, Horace He, Richard Zou and Samantha Andow for help with `functorch` [101] library, Terran Washington, Gopika Jhala and Chantal Mora for designing the Theseus logo, and Oliver Libaw, Christine Gibson, Orialis Valentin, Alyssa Newcomb and Eric Kaplan for help with the blog post. The authors also thank community members for contributions to the open source code. Work by PS and JO was done while at Meta AI.

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
