# Theseus:
# A Library for Differentiable Nonlinear Optimization

## Appendix

## A   Contributions

The contributions of the authors are as follows.

**Luis Pineda** led the engineering of the project, developed and implemented the core API, differentiable nonlinear solvers, motion planning example and tutorials, standard and autodiff cost functions, and backward mode experiments, coordinated with sub-teams to help design, implement, integrate and review of all aspects of the code and evaluations, wrote the paper.

**Taosha Fan** developed and implemented differentiable Lie groups, automatic vectorization, `functorch` support, pose graph optimization example, performance evaluations and benchmarking, helped with API design, bug fixes, did code reviews, wrote the paper.

**Maurizio Monge** developed and implemented all sparse linear solvers, `BaSpaCho` solver and bundle adjustment example, batching, differentiation and custom C++/CUDA backends for sparse solvers, did code reviews, wrote sparse solver and bundle adjustment sections in the paper.

**Shobha Venkataraman** helped develop and implement the motion planning example, helped with API design, wrote several tutorial notebooks.

**Paloma Sodhi** developed and implemented the tactile state estimation example, energy based learning and covariance sampling example, helped design learning API.

**Ricky T. Q. Chen** developed and implemented the direct loss minimization backward mode, wrote its section in the paper.

**Joseph Ortiz** developed and implemented the homography example, helped write its section in the paper, did some bug fixes, helped with the implicit backward mode theory.

**Daniel DeTone** prototyped the homography example, helped develop its final version, wrote its section in the paper.

**Austin Wang** implemented differentiable forward kinematics support, wrote its section in the paper.

**Stuart Anderson** advised on the project, managed and supported research engineers, helped edit the paper.

**Jing Dong** advised on the project, helped design the core API and performance evaluations, did code reviews, helped edit the paper.

**Brandon Amos** advised on the project, developed and implemented implicit and truncated backward modes for nonlinear solvers, helped implement implicit backwards for linear solvers, helped design and analyze backward mode experiments, did code reviews, wrote the backward modes sections and helped edit the paper.

**Mustafa Mukadam** led the project, set the vision and research direction, created and steered the team, provided guidance on all aspects including feature prioritization, API design and implementation, formulated evaluations and experiments, did code reviews, wrote the paper.

## B   Simple example description

In this section, we describe the example in Listing 1 in more detail. The example considers fitting the curve $y = ve^x$ to a dataset of $N$ observations $(x, y) \sim \mathcal{D}$. A standard way to solve this is to minimize the least squares objective Eq. (1) with residuals $r_i(\hat{v}) := y^{(i)} - \hat{v}e^{x^{(i)}}$, for $i = 1, ..., N$, and where $\theta := \hat{v}$. We can model this in `Theseus` with a single `CostFunction` that computes the $N$-dimensional vector $R(\hat{v})$ of all residuals as a function of a single optimization `Variable` $\hat{v}$ and two auxiliary variables, $x$ and $y$.

The code implementing this problem starts by creating uniquely named `Variable` containers in lines 3-5. We then create an objective with the cost function (lines 11-15). We use a `CostFunction` of type `AutoDiffCostFunction` (line 12), which relies on `torch.autograd` and vectorization via `functorch` [101] to automatically compute the residual Jacobians used by the inner optimizer (see App. C). `AutoDiffCostFunction` requires providing an error function that receives optimization variables and auxiliary variables (defined in lines 7-9), and returns `torch` tensors computing the (unweighted) residual. Although not required by this problem, we also illustrate how to add a cost weight to the residuals by including a `ScaleCostWeight`, which simply scales all residuals in this cost function by a scalar (1.0 in this case). Finally, we encapsulate the objective and a Gauss-Newton optimizer into a differentiable `TheseusLayer` in line 16.

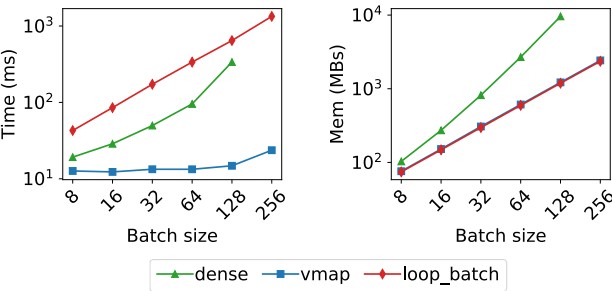

Figure 6: `AutoDiffCostFunction` time (left) and memory (right) consumption in the homography estimation example for different autograd modes and batch sizes.

To illustrate how to differentiate through this layer, we perturb the $x$ values in the dataset so that it becomes impossible to recover the correct value of $v$ from NLS optimization alone, and then define $\phi := x$ and $L(\theta^*(\phi)) := (\hat{v}^*(x) - v)^2$. Thus, the outer loop optimization corrects the $x$ tensor so that the solution of the inner loop matches the true value of $v$.

In the code, the outer parameter is defined in line 18, with initial value for $\phi$ set to a perturbed function of the true $x$, while the outer loss is defined in line 24. In lines 21-23 we solve the NLS problem, by calling `forward()` with the current value of $\phi$ as the value to set for auxiliary variable named "x", and an initial value $v = 1$ for the optimization variable named "v" (with a required batch dimension in the input); we also set the backward mode to `Implicit`. The optimum found can be recovered as a tensor by getting key "v" of the dictionary returned by `forward`, which we then use to compute the outer loss in line 24. Finally, outer loop optimization is done via `torch`'s well-known autograd engine, in lines 25-26, using the Adam optimizer [102] for $\phi$, defined in line 19.

## C `AutoDiffCostFunction` autograd modes

In this section, we evaluate the effect of the three different autograd modes we provide for automatically computing the jacobians for `AutoDiffCostFunction`:

- **dense** uses `torch.autograd.functional.jacobian`, which computes a dense jacobian that includes cross-batch derivatives; i.e., the derivative of $j$-th batch output with respect to variables in the $i$-th batch input. Since we only need per-sample gradients, we slice the result of this operation.

- **loop_batch** is also based on `torch.autograd.functional.jacobian`, but we manually loop over the batch before each call so that we obtain only per-sample gradients.

- **vmap** mode uses `functorch.vmap` [101] to compute per-sample gradients in a vectorized manner.

For evaluation, we use the homography example described in App. E.5, which uses `AutoDiffCostFunction` and can have significant memory requirements when computing the jacobians. Fig. 6 illustrates the advantages of using **vmap** over the other two modes, both in terms of compute time and memory. The **loop_batch** mode has similar memory requirements to **vmap**, but the compute time is significantly slower than the other two methods. Finally, **dense** mode has substantially more memory requirements than the two other methods (up to an order of magnitude higher), and runs out of memory for the largest batch size 256 used in this experiment. For a batch size of 128, **vmap** results in a speedup of 22x over the next best method ($\sim$ 15ms vs. $\sim$ 337ms for **dense**), and almost 8x less memory ($\sim$ 1.2GBs vs. $\sim$ 9.6GBs).

## D Differentiable Lie group details

While differentiation on the Euclidean space is straightforward, it remains challenging to do so on the non-Euclidean manifolds. In this section, we provide details about how to compute the derivatives on the tangent space of Lie groups using the projection operator [58]. The implementation of the projection operator is essential for automatic differentiation on the tangent space of Lie groups.

Suppose $F(g)$ is a function of $g \in G$ where $G$ is a matrix Lie group and $\tau(\xi)$ a retraction map of $G$. For notational simplicity, let $\nabla_g F(g)$ denote the Euclidean gradient of $F(g)$ and $T_e G$ the Lie algebra of $G$. Following [103], the gradient on the tangent space of Lie group is a linear operator $\mathrm{D}_g F(g)$ such that

$$\mathrm{D}_g F(g) \cdot \xi = \left. \frac{\partial}{\partial s} \right|_{s=0} F\big(g\tau(s \cdot \xi)\big) \tag{5}$$

| | Objective Value | | | | | |
|---|---|---|---|---|---|---|
| | Sphere | Torus | Cubicle | Rim | Grid | Garage |
| Initial | $8.437\times10^2$ | $1.234\times10^4$ | $1.622\times10^6$ | $1.924\times10^7$ | $4.365\times10^4$ | $7.108\times10^{-1}$ |
| Final | $6.805\times10^2$ | $1.212\times10^4$ | $1.455\times10^3$ | $4.157\times10^4$ | $4.218\times10^4$ | $6.342\times10^{-1}$ |

Table 1: Initial and final objective values of Theseus on 3D benchmark datasets with PGO example.

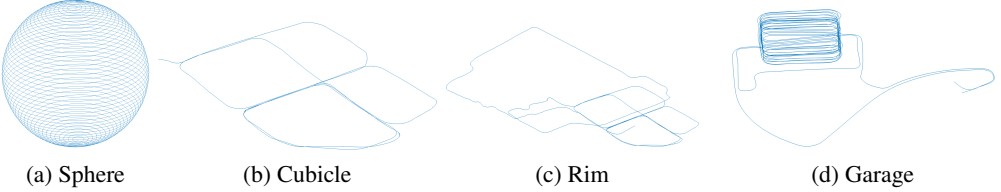

(a) Sphere      (b) Cubicle      (c) Rim      (d) Garage

Figure 7: Qualitative results of Theseus on 3D benchmark datasets with PGO example.

holds for any Lie algebra elements $\xi \in T_e G$. As a result of the chain rule, the right-hand side of the equation above is equivalent to

$$\frac{\partial}{\partial s}\bigg|_{s=0} F\big(g\tau(s\cdot\xi)\big) = \nabla_g F(g) \cdot \frac{\partial}{\partial s}\bigg|_{s=0} g\tau(s\cdot\xi) = \nabla_g F(g) \cdot g\xi \tag{6}$$

where the last equality results from properties of the retraction map. Then, we conclude from Eqs. (5) and (6) that

$$D_g F(g) \cdot \xi = \nabla_g F(g) \cdot g\xi. \tag{7}$$

Therefore, there exists a projection operator $\mathrm{proj}_g(\cdot)$ such that

$$D_g F(g) = \mathrm{proj}_g\big(\nabla_g F(g)\big) \tag{8}$$

for any gradients on the tangent space of Lie group and their corresponding Euclidean gradients [58]. Furthermore, note that the projection operator $\mathrm{proj}_g(\cdot)$ is a linear operator depending on $g \in G$ and can be computed in closed form.

## E    Example application details

### E.1    Pose graph optimization

Pose graph optimization (PGO) [60, 104, 105] is the problem of recovering unknown poses of SE(2) and SE(3) from the noisy relative pose measurements. Pose graph optimization has extensive applications in robotics [35], computer vision [106], computational biology [107], sensor networks [108], etc. In pose graph optimization, we represent unknown poses as vertices and relative pose measurements as edges. Then, it is possible to compute the relative pose errors for each pair of neighboring vertices such that a nonlinear least-squares optimization problem can be formulated for pose estimation. A more detailed introduction to pose graph optimization can be found in [35, 60, 104, 105].

Theseus includes a differentiable and coordinate-independent version of the relative pose errors with which it is straightforward to solve pose graph optimization. We evaluated Theseus on the simulated Cube dataset and a number of benchmark datasets for pose graph optimization [35, 60]. The Cube dataset simulates the 3D odometry of a robot with varying numbers of poses, loop closure probabilities, and loop closure outlier ratios, which is used to profile the time and space complexities of the forward and backward passes in Theseus. Furthermore, the benchmark datasets indicate that Theseus is capable of solving large-scale differentiable nonlinear optimization problems with comparable accuracy and efficiency to existing state-of-the-art solver like Ceres [37].

Theseus and Ceres attain the same objective values for all the evaluated benchmark datasets [35, 60] using the chordal initialization [109]. The inital/final objective values and qualitative results for some benchmark datasets are shown in Table 1 and Fig. 7, respectively.

### E.2    Tactile state estimation

Recent work [28] explored the use of NLS optimization with learned tactile sensor observations for tactile pose estimation. The goal is to incrementally estimate sequences of object poses that are moved by a robotic hand equipped with a DIGIT tactile tensor [61]. The key insight of Sodhi

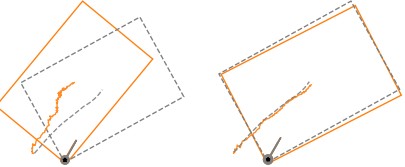 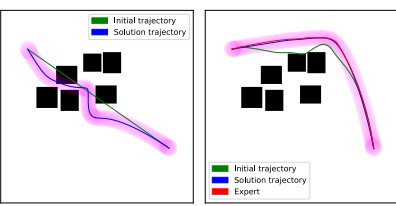

Figure 8: Examples of differentiable tactile state estimation and differentiable motion planning with Theseus. **Left:** Pose estimates before and after learning in the tactile state estimation example. Grey color indicates ground truth, and orange the estimate. The plot shows the trajectory as a curve, and the rectangle indicates the last object pose in the trajectory. **Right:** Trajectories generated by the planner before and after learning. The learned model generates initial trajectories for the optimizer to match an expert with only a few optimization steps.

et al. [28] is to use learning to transform high dimensional tactile observations into relative poses between measurement pairs. Once relative poses are available, the sensor data can be integrated into an optimization problem that solves for object poses. The objective includes four types of cost functions. One penalizes inconsistencies with the measurement coming from the learned observation model. A second one encourages the predicted poses to be consistent with a quasi-static physics model [110]. A third type adds geometric constraints by penalizing intersections between the end effector and the object using a signed distance field. Finally, a fourth cost function incorporates pose priors from a camera.

In Theseus, we implement an offline and differentiable version of the tactile state estimation problem above, using a dataset of 63 trajectories of length 25 with known ground truth poses provided by the authors of [28]; we used 56 of these as training set and the other 7 as a test set. Optimization variables are object and end effector poses (modeled as SE(2) groups) for each point in the trajectory, and the outer loss objective is the difference between the optimized object poses and the ground truth in the dataset. The learnable component corresponds to the relative pose model, using a pre-trained encoder, and finetuning the final layer via end-to-end learning through the inner loop optimization. This approach is similar to how the tactile measurements model was trained in [111], with the two main differences being that we do not use an energy-based formulation and instead directly differentiate through the inner optimizer, and we also do not consider an incremental setting. Fig. 8 (left) shows an example of estimated trajectories before and after learning.

### E.3 Bundle adjustment

Bundle adjustment is the problem of optimizing a 3D reconstruction formed by a set of camera images and a set of landmarks observed and matched across the images. In every camera image a 2D coordinate is identified for the position of all observed landmarks, and the problem is initialized with an estimate of the positions of the landmarks and the camera poses. We call *reprojection error* the image-offset between where the landmark was detected on the image, and the reprojection of the landmark according to the current parameter estimation. The optimization problem consists of simultaneously tweaking the cameras poses and landmark positions, while minimizing the square-sum of all the reprojection errors; see [8, 62] for in depth exploration on bundle adjustment and its state-of-the-art.

We provide a bundle adjustment application example in Theseus, adopting the same data format of [62], with functions to generate synthetic dataset, as well as load/save open source datasets. To test bundle adjustment in a differentiable optimization setting, we add soft-kernels to the reprojection errors and setup as outer loop parameter the radius of the soft-kernel, which represents the confidence radius for reprojection errors with respect to possible outlier observations. We use as outer loss the average frame pose error from a ground truth value, such that the outer loop's task is to set the radius to a value that will make the bundle adjustment problem set the ideal soft loss radius value.

### E.4 Motion planning

NLS optimization can also be used for motion planning in robotics [5], where the objective variables are robot poses and velocities on a set of discrete time steps. Cost functions include terms representing smoothness constraints modeling forward kinematics, collision avoidance penalties, and boundary conditions on start and goal states. An end-to-end differentiable version of this formulation was proposed by Bhardwaj et al. [20], where a neural model predicts state-dependent cost weights for each step in the path, and the outer loss encourages the inner loop optimization to produce paths

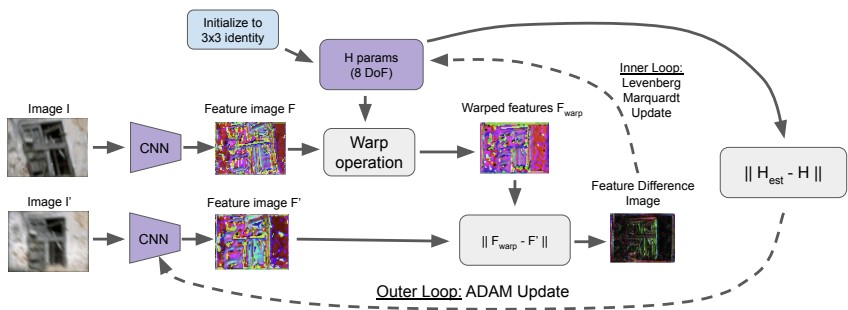

Figure 9: **Learning Robust Image Features for Homography Estimation with Theseus.** An inner loop optimization problem aligns two images via a feature-metric Levenberg-Marquardt optimization using features computed from a CNN. The outer loop uses Adam to update the weights the of the CNN that best minimize the final homography error.

matching an expert in a dataset of trajectories. As part of Theseus, we include differentiable versions of cost functions like smoothness and collision in [5], and an example of how to setup end-to-end differentiation for optimization variable initialization. That is, the model proposes initial trajectories for the optimizer, and the outer loss is set to a weighted sum of two terms, one computing closeness to the expert trajectory, and another equal to the inner loop's objective after only 2 iterations. This loss encourages the model to produce high-quality "proposals" that converge to good quality solutions quickly; an example of before/after training initial trajectories is illustrated in Fig. 8.

### E.5 Homography estimation

A homography, also known as a projective transformation, is a linear mapping between a 2D point in one image $x$ to a point in another image $x'$, defined by a $3 \times 3$ matrix $H$, written as $x \sim Hx'$, where $\sim$ defines the equivalence up to scale. In addition to representing linear transformations across 2D images, the homography is also a valid approximation of the motion of 2D points observed from camera images in 3D scenes in certain scenarios such as (1) rotation-only motion between cameras (2) when the scene is planar and (3) when the scene structure is far from the camera.

One approach to solving for the parameters of the homography is through iterative optimization via dense alignment of RGB image pixels in the image through fast second order optimization methods, as is done in Lucas-Kanade optical flow algorithm [112, 113]. This approach is also known as photo-metric alignment. Though this technique performs well in many scenarios, photo-metric alignment struggles when the lighting in the scene changes significantly, because it assumes that the brightness of a pixel is constant across different views. Feature-metric optimization is an extension to photo-metric optimization that works by first passing the image $I$ through a feature extractor function $f(\cdot)$, such as a convolutional neural network parameterized by weights $w$, $F = f(I; w)$. This function generates a feature map $F \in \mathbb{R}^{C \times H \times W}$, where $C$ is a high dimensional channel number like 32, and $H$ and $W$ represent the image height and width respectively. In feature-metric alignment, the alignment is done at the feature map level, rather than the RGB image level.

One important question when designing a feature-metric optimization algorithm, is how to obtain the weights $w$ that define the feature extractor. One approach used in works such as [63] uses an off-the-shelf CNN which has been trained for image classification. One benefit of using Theseus for such as task is that the learning problem can be written without deriving analytical gradients, making it much easier to rapidly prototype and explore various formulations. In our example, we demonstrate a use-case of Theseus by performing end-to-end training of a two-layer CNN using gradients obtained through the homography optimization. A high level diagram of this learning problem is presented in Fig. 9. We optimize a dense feature-metric mean-squared error term in the inner loop and a four-corner homography error in the outer loop. The four-corner error is a simple measure that computes the L2 distance of four corners of the image after being transformed by the estimated and ground truth homography, as is used in [114] as the output parameterization.

## F    BaSpaCho: Batched Sparse Cholesky

In this section, we provide more details for our open-source novel BaSpaCho solver (https://github.com/facebookresearch/baspacho). BaSpaCho implements the *supernodal* Cholesky algorithm [70] to achieve state-of-the art performance by exploiting dense operations via BLAS/cuBLAS. The heuristics for clustering in the supernodal algorithm evaluate the trade-offs of fragmentation in

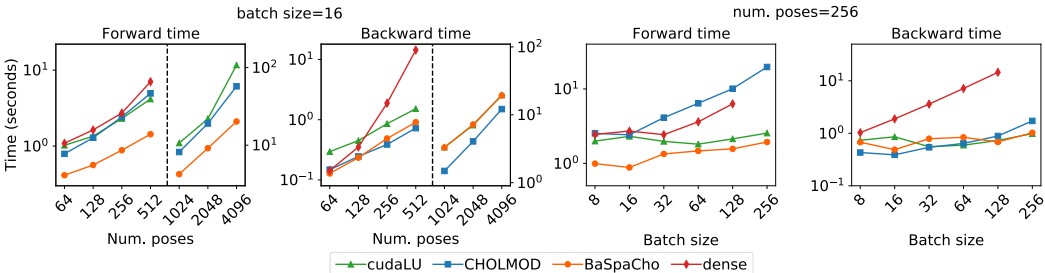

Figure 10: Forward/backward times of `Theseus` with sparse and dense solvers on different PGO problem scales.

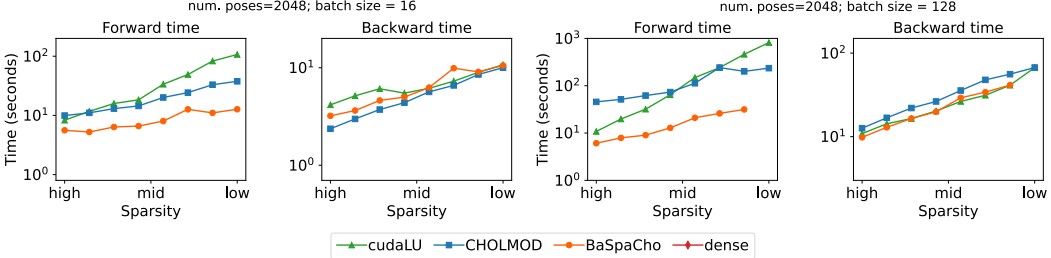

Figure 11: Forward and backward times of `Theseus` with sparse and dense solvers on PGO problems with 2048 poses and different levels of sparsity.

sparse matrices against denser matrices with explicit zero-fill. These heuristics use a computation model that takes into account the architecture (batched/CPU/GPU) that can impact preference towards sparser or denser operations, and allows for further fine-tuning and customization. Apart from a minimal memory allocation needed for the symbolic factorization, BaSpaCho does not own any allocated memory allowing the user to fully manage memory arrays. This allow us to temporarily offload GPU arrays representing factorized matrix data to the CPU when necessary. Unlike existing solvers, BaSpaCho exposes lightweight random accessors that allows the user to read and write matrix blocks in the numeric factor data. This facilitates easy bookkeeping needed by optimization methods which often re-implement block-sparse matrix structures and convert between different matrix formats in order to invoke sparse solvers like CHOLMOD.

## G   Benchmark details and additional results

In this section, we present more profiling results for forward and backward pass of `Theseus`, using the same setup as Sec. 5.1. For evaluation, we used the cube datasets of PGO (see App. E.1) with different numbers of poses, batch sizes, and levels of sparsity. In addition to the forward and backward times as a function of the numbers of poses and batch sizes reported in Sec. 5.1, we further report and analyze the memory usage of `Theseus` with different solvers (cudaLU, CHOLMOD, BaSpaCho and dense) in various settings.

### G.1   Forward and backward times with smaller batch size and number of poses

We profile PGO using different linear solvers (cudaLU, CHOLMOD, BaSpaCho and dense) for fixed batch size of 16 and number of poses of 256. The setup is the same as that in Sec. 5.1 except that fixed batch size and number of poses are smaller. Fig. 10 shows the average time of a full forward and backward pass. Similar to Fig. 3 with larger fixed batch size and number of poses, the sparse solvers are faster than dense. For the smallest problem considered (64 poses, 16 batch size), the total sum of average forward and backward times are 1.32s (cudaLU), 0.94s (CHOLMOD), 0.54s (BaSpaCho), 1.23s (dense) per batch. Increasing to 128 poses makes the sparse solvers noticeably faster than dense: 1.98s (dense) vs 1.80s (cudaLU), 1.53s (CHOLMOD), and 0.80s (BaSpaCho). As the problem scale increases, the gap between the sparse and dense solvers widens: for the largest problem solvable with dense (512 poses, 16 batch size) the average total times are 5.68s (cudaLU), 5.67s (CHOLMOD), 2.34s (BaSpaCho), and 21.37s (dense). The speedup over dense is ∼3.7x for cudaLU and CHOLMOD, and ∼9.1x for BaSpaCho.

### G.2   Forward and backward times with respect to sparsity

We study the forward and backward pass times of `Theseus` with sparse and dense solvers for different levels of sparsity using the synthetic Cube dataset. In PGO, loop closure probability represents how

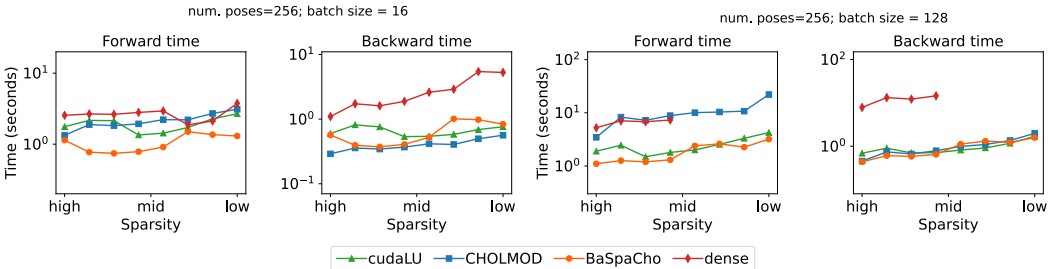

Figure 12: Forward and backward times of `Theseus` with sparse and dense solvers on PGO problems with 256 poses and different levels of sparsity.

likely a pose has a loop closure edge connected to the other poses, and thus, greater loop closure probability yields a less sparse Hessian. We use loop closure probabilities from 0.05 to 0.40 in increments of 0.05, to indicate the level of sparsity for Cube datasets from high (0.05) to low (0.40).

The average forward and backward times of `Theseus` on PGO problems with different levels of sparsity for fixed numbers of poses are shown in Fig. 11 for 2048 poses and in Fig. 12 for 256 poses. In both figures, the left two plots are with 16 batch size and the right two plots are with 128 batch size. As expected, it takes more time in most cases for PGO problems with lower sparsity. Since dense does not exploit the sparsity of optimization problems when solving the linear systems, forward pass of dense takes almost the same amount of time regardless of the levels of sparsity. There is still some overhead for dense as sparsity decreases, because more loop closure edges implies more cost function terms in the objective, so putting together the approximate Hessian is computationally more expensive.

### G.3 Scalability of `Theseus`

In addition to forward and backward times in Sec. 5.1 and Apps. G.1 and G.2, we analyze the scalability of `Theseus` with different linear solvers (cudaLU, `CHOLMOD`, BaSpaCho and dense) following a similar setup to evaluation in Figs. 3 and 10.

We profile PGO with various numbers of poses from 64 to 8192 and batch sizes from 8 to 1024 in increments of power

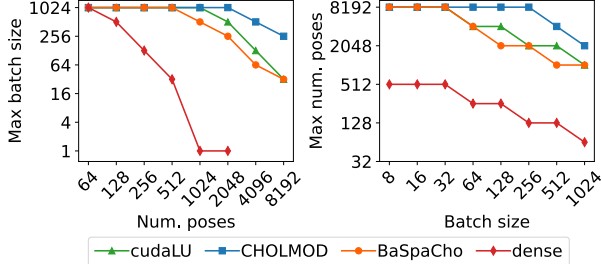

Figure 13: Largest PGO problems `Theseus` scales to for different numbers of poses and batch sizes.

of 2. Fig. 13 shows the maximum batch sizes solvable for given numbers of poses (left) and the maximum numbers of poses solvable for given batch sizes (right). In Fig. 13, it can be seen that dense only scales to small PGO problems due to the memory limitation and fails to solve any PGO problems with 4096 poses or more, even with batch size of 1. In contrast, cudaLU, `CHOLMOD` and BaSpaCho successfully solve PGO problems with 8192 poses for a batch size of 32 (cudaLU, BaSpaCho) and 256 (`CHOLMOD`). As discussed in Sec. 5.1, cudaLU and BaSpaCho require extra GPU memory to solve linear systems, whereas `CHOLMOD` has all computation run on CPU, and thus can solve larger DNLS problems than cudaLU and BaSpaCho.

### G.4 Comparison with `Ceres` on smaller batch size and number of poses

In addition to Sec. 5.2, we follow the same setup to compare `Theseus` as a stand-alone NLS optimizer with `Ceres` for PGO problems with a smaller fixed batch size of 16 and number of poses of 256. Fig. 14 shows the speedup of `Theseus` compared to `Ceres` (black dotted line). Similar to Fig. 4, `Ceres` is faster for PGO problems for small batch sizes and numbers of poses, and `Theseus` is faster as the problem scale increases.

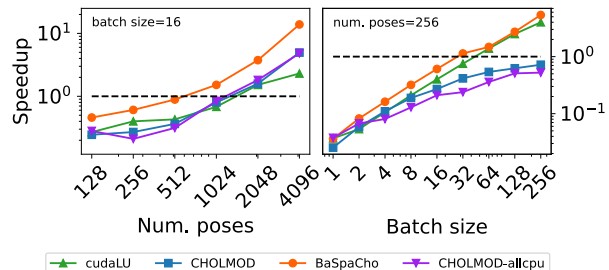

Figure 14: Speedup of `Theseus` (forward pass) over Ceres (black dashed) on different PGO problem scales.

# H    Backward mode details, additional results, and derivations

## H.1    Experimental details

In Sec. 5.3 we use the tactile state estimation example to evaluate the performance of different backward modes. As mentioned in App. E.2, the dataset consists of 63 trajectories of length 25, 56 of which we use for training and 7 for test. We use a batch size of 8 and train for 100 epochs, resulting in 700 batches for averaging time and memory results. For the inner loop, we use Gauss-Newton with a step size of 0.05; in the test set we run the inner loop for 50 iterations, regardless of the number used during training. For the outer loop, we use the Adam optimizer with a learning rate of $10^{-4}$, decayed exponentially by a factor of 0.98 after every epoch.

## H.2    Additional results

Fig. 15 (left) shows the peak memory consumption during the forward pass. We observe the same trend from the backward pass (Fig. 5, center right), where Unroll's memory consumption increases linearly with the number of inner loop iterations, while for the other methods it remains constant. Implicit has the lowest peak memory requirement ($\sim$22MBs), followed by DLM ($\sim$29MBs).

Fig. 15 (center, right) also shows training curves for all methods. We observe that, despite higher performance in the test set, Implicit is more unstable during training and oscillates between low and high values; this suggests that careful use of early stopping and hyperparameter tuning might be required when using Implicit. The other methods are more stable, with the two truncated methods achieving the lowest training loss after Implicit. Fig. 15 (right) shows that Unroll's performance degradation, relative to other methods, with increasing number of inner loop iterations (also see Fig. 5, right) is not just a generalization issue, but also happens during training. This suggest possible numerical issues from unrolling gradients over a high number of optimization steps, as observed in prior work.

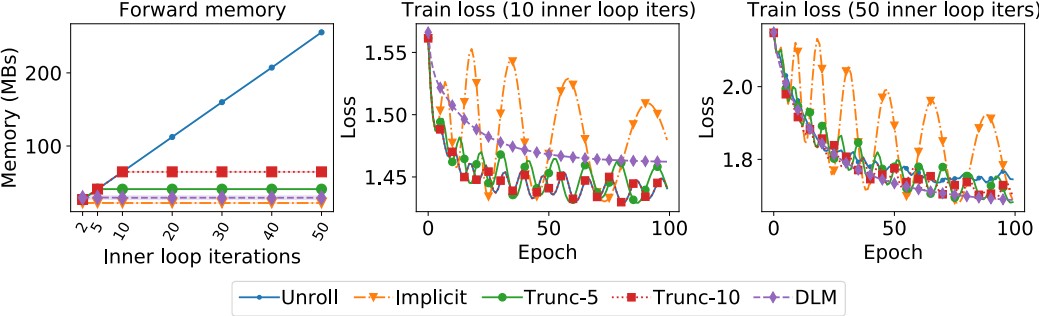

Figure 15: Additional results for backward modes comparison in tactile state estimation problem. **Left:** Memory consumption in forward pass. **Center:** Training loss when using 10 inner loop iterations. **Right:** Training loss when using 50 inner loop iterations.

## H.3    Backward modes summary and limitations

Fig. 16 visualizes the backward modes and Table 2 contrasts their limitations. The table shows that all four modes can be used when learning parameters for cost functions or cost weights. However, unlike other approaches, Implicit cannot be used when learning initial values for the optimization variables, $\theta_{\text{init}}$. Another limitation of Implicit is that the resulting gradients might be inaccurate in problems where it is not feasible to find the optimal solution to the inner optimization problem; other methods don't experience this limitation, since they compute gradients around the approximate solution found. On the other hand, both Unroll and Trunc could potentially experience vanishing or exploding gradient issues when the number of iterations to backpropagate through is large, a limitation that is not shared by Implicit and DLM. Finally, a limitation of DLM is that $\epsilon$ needs to be tuned (see Eq. (16)), which can greatly affect performance. Likewise, the number of backward iterations for Trunc may also require some tuning.

## H.4    Derivations for backward modes

### H.4.1    Implicit function theorem

For adjoint differentiation, we make use of the implicit function theorem, which is originally from Dini [115], and presented in Dontchev and Rockafellar [39, Theorem 1B.1] as:

|  | Cannot be used for learning $c_i$ or $w_i$ | Cannot be used for learning $\theta_{\text{init}}$ | Requires accurate $\theta^*$ solution | Possible vanishing or exploding gradients | Requires tuning | Compute and memory usage |
|---|---|---|---|---|---|---|
| Unroll |  |  |  | ✓ |  | high |
| Trunc |  |  |  | ✓ | ✓ | medium |
| Implicit |  | ✓ | ✓ |  |  | low |
| DLM |  |  |  |  | ✓ | low |

Table 2: Backward modes summary and limitations.

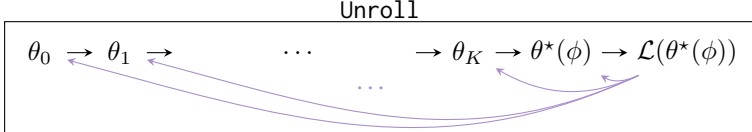

Figure 16: Illustration of the dependencies of the backward modes for computing $\nabla_\phi \mathcal{L}(\theta^\star)$.

**Theorem 1** (Dini's implicit function theorem). *Let the roots of $g(\theta; \phi)$ define an implicit mapping $\Theta^\star(\phi)$ given by $\Theta^\star(\phi) := \{\theta \mid g(\theta; \phi) = 0\}$, where $\theta \in \mathbb{R}^m$, $\phi \in \mathbb{R}^n$, and $g : \mathbb{R}^m \times \mathbb{R}^n \to \mathbb{R}^m$. Let $g$ be continuously differentiable in a neighborhood of $(\bar{\theta}, \bar{\phi})$ such that $g(\bar{\theta}; \bar{\phi}) = 0$, and let the Jacobian of $g$ with respect to $\theta$ at $(\bar{\theta}, \bar{\phi})$, i.e. $\mathrm{D}_\theta g(\bar{\theta}; \bar{\phi})$, be non-singular. Then $\Theta^\star$ has a single-valued localization $\theta^\star$ around $\bar{\phi}$ for $\bar{\theta}$ which is continuously differentiable in a neighborhood $Q$ of $\bar{\phi}$ with Jacobian satisfying*

$$\mathrm{D}_\phi \theta^\star(\tilde{\phi}) = -\mathrm{D}_\theta^{-1} g(\theta^\star(\tilde{\phi}); \tilde{\phi}) \mathrm{D}_\phi g(\theta^\star(\tilde{\phi}); \tilde{\phi}) \qquad \text{for every } \tilde{\phi} \in Q. \tag{9}$$

### H.4.2  Proof of Prop. 1

*Proof.* Let $\bar{\phi}$ be a hyper-parameter resulting in a unique $\theta^\star(\bar{\phi})$ and recall that the implicit mapping for Eq. (1) is defined by $g(\theta; \phi) := \nabla_\theta S(\theta, \phi)$ and is zero at the optimal parameters, i.e. $g(\theta^\star(\bar{\phi}), \bar{\phi}) = 0$. Let $h(\theta; \phi) := \theta - [\nabla_\theta^2 S(\theta; \phi)]_{\text{stop}}^{-1} \nabla_\theta S(\theta; \phi)$ be the Newton iteration where $[\cdot]_{\text{stop}}$ is a function that zeros the derivative. Differentiating $h$, which can be done using automatic differentiation on a Newton step, results in the implicit derivative Eq. (9):

$$
\begin{aligned}
\mathrm{D}_\phi h(\theta^\star(\bar{\phi}); \bar{\phi}) &= \mathrm{D}_\phi \left( \theta - \left[\nabla_\theta^2 S(\theta^\star(\bar{\phi}); \bar{\phi})\right]_{\text{stop}}^{-1} \nabla_\theta S(\theta^\star(\bar{\phi}); \bar{\phi}) \right) \\
&= - \left[\nabla_\theta^2 S(\theta^\star(\bar{\phi}); \bar{\phi})\right]_{\text{stop}}^{-1} \mathrm{D}_\phi \nabla_\theta S(\theta^\star(\bar{\phi}); \bar{\phi}) \\
&= -\mathrm{D}_\theta^{-1} g(\theta^\star(\bar{\phi}); \bar{\phi}) \mathrm{D}_\phi g(\theta^\star(\bar{\phi}); \bar{\phi})
\end{aligned}
\tag{10}
$$

$\square$

### H.4.3  Direct loss minimization for use in Theseus

Originally, DLM gradient for non-linear objective functions [41] can be expressed as

$$\nabla_\phi L = \lim_{\varepsilon \to 0} g_{\text{DLM}}^\varepsilon, \quad \text{where} \quad g_{\text{DLM}}^\varepsilon \triangleq \frac{1}{\varepsilon} \left[ \frac{\partial}{\partial \phi} S(\theta^*; \phi) - \frac{\partial}{\partial \phi} S(\theta_{\text{direct}}; \phi) \right] \tag{11}$$

where

$$\theta^* = \arg\min_{\hat{\theta}} S(\hat{\theta}; \phi), \qquad \text{and} \qquad \theta_{\text{direct}} = \arg\min_{\hat{\theta}} S(\hat{\theta}; \phi) - \varepsilon L(\hat{\theta}). \qquad (12)$$

However, this dependence on the loss function fits poorly in a reverse-mode automatic differentiation framework like `PyTorch`. Instead, we can construct an equivalent formulation by noting that in continuous space, we can first linearize the loss function around the current solution $\theta^*$,

$$\hat{L}(\theta) = L(\theta^*) + \nabla_\theta L(\theta^*)(\theta - \theta^*) \qquad (13)$$

Let $v = \nabla_\theta L(\theta^*)$, then the perturbed solution becomes

$$\theta_{\text{direct}} = \arg\min_{\hat{\theta}} S(\hat{\theta}; \phi) - \varepsilon\left(L(\theta^*) + v^T(\hat{\theta} - \theta^*)\right) = \arg\min_{\hat{\theta}} S(\hat{\theta}; \phi) - \varepsilon v^T \hat{\theta}. \qquad (14)$$

Plugging this back into Eq. (11), we see that this is an algorithm which takes in a gradient vector $v$ and computes an approximation to the vector-Jacobian product $\nabla_\phi L(\theta^*) = v \frac{\partial \theta^*}{\partial \phi}$.

As `Theseus` is designed to solve optimization problems where $S$ is expressed as sum of squares, it cannot readily handle solving $\theta_{direct}$ as this requires adding a linear term to the objective. Instead, let us consider the following "completing the square" approach:

$$\arg\min_{\hat{\theta}} \|\varepsilon\hat{\theta}\|^2 - \varepsilon v^T \theta = \arg\min_{\hat{\theta}} \varepsilon^2 \hat{\theta}^T \hat{\theta} - \varepsilon v^T \hat{\theta} + \left(\tfrac{1}{2}v\right)^T \left(\tfrac{1}{2}v\right) = \arg\min_{\hat{\theta}} \left\|\varepsilon\hat{\theta} - \tfrac{1}{2}v\right\|^2 \qquad (15)$$

We can thus add this extra term and let

$$\theta_{\text{direct}} = \arg\min_{\hat{\theta}} S(\hat{\theta}; \phi) + \left\|\varepsilon\hat{\theta} - \tfrac{1}{2}v\right\|^2 \qquad (16)$$

This adds a small bias to the gradient due to the addition of $\|\varepsilon\hat{\theta}\|^2$ but when $\varepsilon$ is small it shouldn't be problematic. In practice, we solve for $\theta_{\text{direct}}$ by starting from $\theta^*$ and using just one iteration of Gauss-Newton.