# OpenReview forum: "Theseus: A Library for Differentiable Nonlinear Optimization"
_NeurIPS.cc/2022/Conference — NeurIPS 2022 Accept_

### Official Review · Reviewer_aWKw · 2022-07-08

**Rating:** 6
**Confidence:** 3
**Soundness:** 3 good
**Presentation:** 3 good
**Contribution:** 3 good

**Summary:**

This paper describes a convenient, systematic, and clean DNLS library. It could be possibly used to develop many differentiable algorithms.

It has several useful features, such as Differentiable Lie groups and kinematics.

Although it’s not novel, this feature makes optimization of matrices and robots kinematics more easily in pytorch. It could be useful when optimizing non-Euclidean variables, for example, joints in robotics, pose matrices in multi-view geometry like SLAM and SfM.

Automatic Vectorization: It could possibly be one novelty, but was not described in detail.  In L232, “we make use of the single instruction-multiple-data (SIMD) protocol by automatically detecting and vectorizing operations of the same type, significantly reducing the overhead for forward and backward passes.”

GPU Acceleration for a Solver:  They design and implement a differentiable cudaLU solver, which itself could be useful. Cuda solver is usually closed-source. Making a differentiable version of it takes effort.

 Four Differentiation Strategies:  In Sec 4.3, they provide 4 different strategies for users, suitable for different problems. It helps the user to select the most suitable one.

**Questions:**

Can you highlight or give more detail on the  key novelties of the design choices & algorithms in the library itself?

**Limitations:**

Not discussed clearly.

**Strengths And Weaknesses:**

STRENGTH:  potentially high utility for the ML community interested in differentiable programming paradigm

With this library, researchers can more easily design differentiable programming systems while spending less time on the differentiation part (or at least can quickly code up "proof of concept" systems using this general tool), esp. considering that the optimization is one of the hardest parts when making a system differentiable.

WEAKNESS:  novelty in this paper is unclear, if not low, as it is currently presented and described.

Another possible weakness is that, similarly to other methods that achieve differentiation by pytorch, the overhead could be high compared to simple forward C++ code, especially when the data size is small and the advantage of parallelization is not significant. An example is in Figure 3, where the existing Ceres (black dashed) outperforms most of the methods in this new library on small data.

---

> ### Author Response · Authors · 2022-08-01
> **Response to Reviewer aWKw**
>
> We thank the reviewer for their feedback. We are pleased to see that the reviewer thinks that with this library researchers “can more easily design differentiable programming systems while spending less time...”; this is one of our main goals! We are also glad the reviewer recognized some of the engineering challenges we tackled: “Cuda solver is usually closed-source. Making a differentiable version of it takes effort.”
>
> The reviewer points out that the novelty of this paper is unclear, to which we respectfully disagree. The last three paragraphs of the introduction in the paper state our novel contributions in detail. Theseus is essentially **the first library** that offers a “full-stack” solution for differentiable NLS including sparse solvers, Lie-algebra support, a wide range of out-of-the-box cost functions for robotics and computer vision, support for differentiable robot kinematics, support for custom cost functions and an easy to use API. As other reviewers have already pointed out, there is value for the community in having such a library, which did not exist before Theseus.
>
> We agree with the reviewer that there will be overhead compared with forward code written in C++, but we must stress that our purpose is to offer a flexible library for research in *differentiable* optimization, most of which is written in Python, and a significant portion of it in PyTorch. As such, directly comparing computation against libraries that don’t support differentiation would be unfair. In the paper, we have included a benchmark against Ceres to both validate the correctness of our implementation, and to show that we have put a lot of emphasis in performance. We think the fact that Theseus can outperform Ceres in the batched setting for medium-to-large problems shows that performance is a major consideration. But if the user only needs to do non-differentiable forward optimization, this is not the problem that Theseus was designed for.
>
> We also refer the reviewer to Sec 6 in the paper where we discuss limitations in detail which is also acknowledged by the other reviews as “properly discussed” and “clearly addressed”.

---

### Official Review · Reviewer_WkJ4 · 2022-07-09

**Rating:** 8
**Confidence:** 3
**Soundness:** 3 good
**Presentation:** 3 good
**Contribution:** 4 excellent

**Summary:**

The paper presents Theseus, a library for optimizing through differentiable nonlinear least squares. The library allows for end-to-end learning for various structured learning problems like pose estimation, and homography estimation among others. The library is task agnostic, and provides an efficient implementation for dense and sparse problems, along with a variety of optional backprop methods for the underlying second-order optimizer.

**Questions:**

1. Reiterating from above, I would like to see comparisons with existing task specific implementations to see differences if any.
2. In terms of the optimization, the approach builds on a two step optimization scheme. However, this may lead to instabilities in optimization especially with the choice of wrong hyperparams. How much does the choice of hyperparameters affect the overall performance of the proposed approach?

**Limitations:**

The authors have been upfront about the limitations and have clearly addressed them in the settings.

**Strengths And Weaknesses:**

**Strengths**:
1. The proposed library is task-agnostic allowing for end-to-end learning across a variety of robotics and vision problems.
2. The implementation supports a variety of backpropagation approaches for taking gradients including autodiff.
3. The paper shows either comparable or better results as compared to Ceres solver.
4. The motivation is well supported with several examples and has value to the community.


**Weaknesses**
1. Technically, there do not seem to be any flaws in the proposed approach.
2. The authors cite several papers that already implement related approaches. It will be good to compare a general purpose solver like Theseus with application specific implementations.
3. The writing needs

---

> ### Author Response · Authors · 2022-08-01
> **Response to Reviewer WkJ4**
>
> We thank the reviewer for their thoughtful feedback. We are glad that the reviewer highlights the task-agnostic nature of our library and that its “motivation is well supported... and has value to the community” .
>
> Regarding comparing with application specific implementations, in this paper we put a lot of effort in evaluating and demonstrating the correctness of our differentiable optimizer, by comparing our results with an established stand-alone optimizer (Ceres), performing thorough unit testing of all gradient computations, and adding diverse illustrative examples of how to do learning through a TheseusLayer. We hope that this offers enough evidence that our features are sound and useful. We thus argue that comparing with application specific implementations is outside the scope of the paper, considering that a lot of their performance depends on the neural architecture used, which can be quite complicated and might require extensive hyperparameter tuning. We have opted for focusing our efforts on evaluating features that are direct responsibility of our library.
>
> Regarding the impact of hyperparameters for differentiable NLS optimization, this is likely to be problem dependent, and in many cases it can have a significant impact. We have thus added a lot of flexibility for users to easily change hyperparameters for the inner optimization when experimenting in their own research applications. This includes, number of iterations, choice of optimizer, choice of backward method, tolerance, turn on/off Lie-group gradient projection, among others. We hope that researchers find this useful in the future, and plan to adapt these options with evolving needs of the library’s users.

---

### Official Review · Reviewer_nzsL · 2022-07-10

**Rating:** 8
**Confidence:** 2
**Soundness:** 4 excellent
**Presentation:** 3 good
**Contribution:** 4 excellent

**Summary:**

This paper presents Theseus, a software library allowing for a new layer in pytorch models based on differentiable nonlinear least squares (DNLS) solves.  That is, the input to the layer specifies a NLS problem which in the forward pass the layer solves and in the backward pass, the layer provides the adjoint derivates, i.e. derivatives for the solution to the NLS problem in terms of the problem parameters.  This creates a nested optimization problem in which an inner NLS solver iterates to find the NLS solution as part of the outer optimization of a pytorch model by gradient descent.

The authors are clear that they are not introducing this technique but rather provide an extensive list of citations for its use in robotics and vision communities for a solving an array of problems in which combining the flexibility of the deep learning with the constrained structure of NLS leads to good solutions.  However, the authors of this work note that in other works this problem was solved using custom application specific implementations which were in many cases inefficient.  They remedy the problem by providing an efficient and scalable package which can work for many applications.

The paper contains many examples of use cases for the method, explanations of the many options provided for both the forward and backward modes, and empirical profiling comparing the different options and showing improved efficiency versus the previous NLS solver at large scale.

**Questions:**

## Minor Point
- Line 74: $\in$ should be $\subset$.  Do the subsets need to be disjoint?
- Line 76: parallelism issue.  ``Euclidean vector or a matrix Lie group element''
- Line 91: maybe a small semantic point, but this does not really seem like a second-order method since the 2nd derivatives do not need to be computed, which seems like an advantage.  I have seen the derivation and I understand that $J^T J$ is the second order term of the objective after linearizing the model, but it seems one should be careful to be precise here.
- Line 106: "the lastest"

## Questions
- Figure 1: It seems the larger batch has smaller speed up.   Why do you think that is?

## Post-Response Edit
I have read the other reviews and author responses.  These strengthen my positive assessment and I maintain my score.

**Limitations:**

The paper discusses limitations of various alternatives throughout the work, for example, the fact that the CHOLMOD spare solver is tied to the CPU.  A limitations section at the end lays out other limitations, for example, incompatibility with hard constraints.

**Strengths And Weaknesses:**

## Strengths

- The authors make a compelling case for the usefulness of the method implimented by their package, giving an extensive bibliography of uses in robotics and vision.
- They also implement two novel (in this context) methods for computing the adjoint derivatives for DNLS: Implicit differentiation and Direct Loss minimization.  Experiments in Sec. 5.1 show an improvement in efficiency and accuracy relative to unrolling or truncated differentiation.
- The library is very flexible providing two different sparse NLS solvers: CHOLMOD and cudaLU, and explaining the advantages and drawbacks of each.  Four methods for the backward mode are provided.  There is also support for optimizing variables in Lie groups such as orientations in SO(3) which are quite important in robotics and vision and tricky to optimize over.
- Support for batching and automatic vectorization gives the DNLS layer similar levels of parallelism to other pytorch layers.
- The supplementary material includes the complete code of the package including scripts and instructions to run several different examples in the paper.  The appendix includes nice explanations of the examples and proofs justifying the novel backward modes.  Given that the purpose of the paper is provide a solution which works flexibly across different applications, I found seeing these examples script and explanations helpful to understanding this.

## Weaknessess
- Some of the empirical evaluations are somewhat hard to interpret.  Figure 2 left is not a very clear plot and it is hard to find a strong conclusion of when to prefer either sparse solver (although the preference over dense is clear.)
- I understand that it is a prior assumption from the literature that the combination of deep learning and NLS afforded by this method is a worthwhile and superior approach in many applications relative to either DL or NLS independently.  It still would have been convenient if this comparison could be made explicitly for one of the examples.
- Figure 4, right shows that the choice of backward method inside Theseus have an impact on the ultimate performance of the model.  The authors note this is probably application dependent.  This seems like a critical aspect for end users and should probably be explored more.

---

> ### Author Response · Authors · 2022-08-01
> **Response to Reviewer nzsL**
>
> We thank the reviewer for their comprehensive feedback. We are pleased to see that the reviewer found our library flexible, mentioning features such as sparse solvers and SO(3) support, that they see novelty in our adjoint differentiation methods in this context, and that they found our examples helpful in demonstrating the flexibility of the library.
>
> The reviewer has pointed out that Figure 2 is somewhat hard to interpret and to derive a strong conclusion from. We agree that the explanatory text can be improved and plan to do so in the final revision. The main takeaway is that both solvers are comparable in terms of running time, with perhaps a slight advantage of LUCuda solver for smaller problems; moreover, if not many CPU-threads are available for CHOLMOD, any advantage of cudaLU is likely to increase. Thus, for most cases, starting with cudaLU is a good rule of thumb. On the other hand, as mentioned in Appendix D.2, for very large problems and GPU-memory constraints the CHOLMOD-based solver might be a better choice, as the solver for the most part runs on CPU and is thus less memory constrained.
>
> We agree with the reviewer that it would be convenient to see the advantage of structured end-to-end learning with DNLS. However, we argue that this is outside the scope of the paper given that such advantages have already been illustrated in many previous work in this area (many of which we cite, as the reviewer pointed out), and it thus would distract from the focus of the paper, which is describing our flexibility- and efficiency-based design, and benchmarking our different features.
>
> Similarly, we agree with the reviewer that exploring different backward modes across many applications would be very interesting, and it’s indeed one of the reasons why we have created the library: so that researchers can do this more easily in the future. Doing a comprehensive analysis of this across many applications is nevertheless outside the scope of this paper.
>
> Regarding the minor questions, thanks for pointing these out, we will fix them in the final version of the paper.  The sets in line 74 do not have to be disjoint. Regarding the smaller speed up with larger batch size (Fig. 1), note that in this experiment we only run a single batch and that GPU computation generally slows down with increasing batch size. Since vectorization has a multiplicative effect on the batch size, at some point the cost of having a very large batch starts outweighing the savings resulting from parallel processing. We will also clarify that the Hessian computed in NLS is approximate ($J^TJ$) unlike second order Newton‘s method.

---

### Official Review · Reviewer_A8gP · 2022-07-12

**Rating:** 7
**Confidence:** 3
**Soundness:** 4 excellent
**Presentation:** 4 excellent
**Contribution:** 4 excellent

**Summary:**

The paper introduces a flexible, efficient, and task-agnostic open source library for nonlinear least squares problems. By integrating with PyTorch, the package allows one to learn and solve many tasks in computer vision and robotics in an end-to-end and deep fashion. The library provides support for common objectives, robust functions, optimizers (*e.g.*, GN, LM), as well as useful utilities (*e.g.*, Lie-algebra). The underlying implementation is carefully designed such that it is memory and computationally efficient.


**Questions:**

Please see the above weakness section.

**Limitations:**

The limitations are properly discussed.

**Strengths And Weaknesses:**

**Strength**
- I am personally a big fan and a practitioner of deep structure models. Many of my previous work have combined deep neural networks with optimization (some are NLS while some are not). In order to train the whole model end to end, I had to write all these things myself, such as unrolling GN step, lie-algebra transformation, etc. It was fun, but indeed a bit cumbersome. It is thus very exciting to see that there is a flexible and modularized library that can greatly improve the speed of our future development. I can foresee the library being adopted by a wide variety of the community.
- The authors provides several case studies (which is just the tip of the iceberg) across computer vision and robotics and showcase how they can be implemented easily and cleanly with the task-agnostic interface.
- Theseus provide additional support/integration for sparse linear solvers, which can be used independently as standalone pytorch functions.
- Thorough analysis on computational cost and memory usage.
- I also like how the library integrates many (latest) techniques, such as projection operator for Lie-algebra, implicit differentiation, etc. It allows one to experiment with more crazy stuff such as unrolling tons of steps, or obtain more stable results with gradient explosion.


**(Minor) Weakness**
- While the aim of the package is to support generic nonlinear optimization, the current state seems to focus majorly on least square problem (*e.g.* GN solver). The title is thus a bit mis-leading imo.
- Following above, many state-of-the-art algorithms in vision and robotics adopt LM solver with learnable, data-driven $\lambda$. I don't see why this function is unavailable, or what's the major difficulty/block on this.

**Correctness**
- I'm not aware of any correctness issue in this paper.


**Minor**
- I personally gave it a try and use Theseus to replace my personal implementation on GN solver and lie-algebra (for 3D pose estimation). While I didn't observe significant speedup (which aligns with the analyses in the paper), the code is indeed much cleaner ;)

---

> ### Author Response · Authors · 2022-08-01
> **Response to Reviewer A8gP**
>
> We thank the reviewer for their thoughtful feedback. We are glad that they found our library “very exciting” and with potential to be adopted by a wide variety of the community. We are also pleased that the reviewer sees value in our supported features like sparse solvers, projection operators for Lie-algebra, and implicit differentiation. Moreover, it was exciting to see that the reviewer tried out the code for their own problem and found it helpful, even if not significantly faster for their problem.
>
> The reviewer pointed out that Theseus doesn’t currently support learnable data-driven $\lambda$ for LM solver, which is correct. We agree with the reviewer that this feature should be relatively easy to support, and it’s indeed on our stack of coming features. For our dense solvers, adding this is straightforward, by simply allowing $\lambda$ to be a PyTorch tensor; however, adding this is a bit tricker with our custom sparse solvers, and would require some non-trivial amount of work. Thus, the main reason for its current omission has been limited time during which we have prioritized other features.

---

### Meta-Review · Area_Chair_6MBC · 2022-08-31

**Recommendation:** Accept
**Confidence:** Certain

**Metareview:**

This paper presents Theseus, a software library which provides a new layer in the form of a differentiable nonlinear least squares (DNLS) solver. Forward pass solves the problem and the backward pass provides derivates for the optimum with respect to parameters. The reviewers uniformly appreciated the presentation of the paper and the usefulness and packaging of the proposed library. The library's features such as sparse solvers and Lie group algebra was also appreciated by the reviewers. Overall the paper and the library are a strong contribution to the Neurips community and hence I am happy to recommend for acceptance. I would urge the authors to make sure they follow through on feature requests and other suggestions made by the reviewers.

**Award:**

No

---

### Decision · Program_Chairs · 2022-09-14

Accept